# Co-Evolving Latent Action World Models

## Abstract

Adapting pre-trained video generation models into controllable world models via *latent actions* is a promising step towards creating generalist world models. The dominant paradigm adopts a two-stage approach that trains latent action model (LAM) and the world model separately, resulting in redundant training and limiting their potential for co-adaptation. A conceptually simple and appealing idea is to directly replace the forward dynamic model in LAM with a powerful world model and training them jointly, but it is non-trivial and prone to representational collapse. In this work, we propose **CoLA-World**, which for the first time successfully realizes this synergistic paradigm, resolving the core challenge in joint learning through a critical warm-up phase that effectively aligns the representations of the from-scratch LAM with the pre-trained world model. This unlocks a co-evolution cycle: the world model acts as a knowledgeable tutor, providing gradients to shape a high-quality LAM, while the LAM offers a more precise and adaptable control interface to the world model. Empirically, CoLA-World matches or outperforms prior two-stage methods in both video simulation quality and downstream visual planning, establishing a robust and efficient new paradigm for the field.

## 1 Introduction

A prevailing goal in artificial intelligence is the creation of a generalist agent capable of acting across a multitude of environments and embodiments. Central to this vision is the concept of a *world model* (Sutton, 1990; Ha & Schmidhuber, 2018), an internal simulator of the environment that allows an agent to plan and learn through imagination. An ideal world model would be universal, leveraging vast priors about world physics and dynamics, and adaptable with minimal data to any specific downstream task. While large-scale video generative models (OpenAI, 2024; Blattmann et al., 2023) have emerged as powerful candidates for such general-purpose simulators due to their rich pre-trained knowledge, a fundamental challenge remains: how to interactively control the generation. The heterogeneity of action spaces across different domains, from the continuous torques of a robot arm to the discrete button presses of a game console, prohibits the direct use of real actions for finetuning a video generative model to a single, universal world model.

To bridge this gap, Latent Action Models (LAMs) have shown great promise (Schmidt & Jiang, 2023; Bruce et al., 2024; Ye et al., 2025). By inferring abstract actions directly from visual observations, LAMs provide a unified, embodiment-agnostic interface for controlling a world model. This paradigm opens an exciting direction: pre-training a single, general-purpose world model conditioned on a universal latent action space (Bruce et al., 2024; NVIDIA et al., 2025; Gao et al., 2025). To integrate LAMs with world models, existing works typically adopt a two-stage approach: first training a LAM on action-free videos, usually with a small inverse dynamics model (IDM) and a forward dynamics model (FDM) trained from scratch, and then freezing the IDM to supply latent actions for training a larger world model.

However, this two-stage approach faces several issues. First, the FDM and the world model are essentially both performing next-observation prediction, rendering the overall framework redundant. Second, the pipeline forces the world model to rely on a fixed, static latent action space, preventing the latent actions from adapting as world model training progresses. One question naturally arises:

*Can we replace the FDM with the world model?*

At first glance, this might seem like a straightforward modification, but our experiments show that naively training the IDM and world model together can easily lead to collapse.

In this work, we explore this question and provide an affirmative answer. We propose **CoLA-World**, a training pipeline that enables the synergistic co-evolution of latent action learning and world modeling. We first observe that, whether the IDM is initialized from scratch or from a pre-trained one, direct joint training with the world model leads to collapse. This suggests that the IDM is not well aligned with the pre-trained weights of the world model.

To address this, before switching to joint training, CoLA-World introduces a warm-up phase in which the world model is kept frozen and only supplies gradients to update the IDM. This greatly stabilizes subsequent joint training and enables the IDM and world model to co-evolve effectively. On one hand, the powerful world model carries prior knowledge of plausible physics and visual dynamics inherited from a pre-trained video generation model. It acts as an active tutor, providing gradients that guide the from-scratch IDM toward higher-quality latent actions. On the other hand, as the IDM learns to produce a more informative latent action space, it in turn offers the world model a clearer and more precise control interface.

We evaluate our method on a large-scale dataset consisting of human egocentric and robot manipulation videos. Compared to baseline two-stage methods, CoLA-World learns higher-quality latent actions and achieves stronger world model prediction performance. We further provide empirical evidence that co-evolution in the joint-training phase is crucial, as it enables both latent action learning and world modeling to outperform setups where either component is fixed. Finally, we assess the adaptability of the learned latent-action-based world models to out-of-distribution real-action control interfaces, showing that the joint training enabled by our method is key to improving both video prediction quality and downstream visual planning.

In summary, our main contributions are:

- We propose CoLA-World, the first framework that successfully enables joint training of a latent action model with a pre-trained video-generation-based world model.

- Compared to prior two-stage methods, CoLA-World's joint latent action learning and world modeling yield a higher-quality latent action space and a world model with stronger controllability and sample efficiency, improving both video simulation and downstream visual planning.

- We show that CoLA-World's joint training exhibits synergistic co-evolution: the improving world model and LAM mutually reinforce each other, creating a tightly coupled system that drives superior adaptability.

## 2 RELATED WORK

**Latent Action Learning** Latent actions have recently emerged as a promising approach for behavior pre-training on action-free data. Early methods such as FICC (Ye et al., 2023) and LAPO (Schmidt & Jiang, 2023) adopt the IDM–FDM framework, where latent actions are discovered through a next-frame reconstruction objective. Genie (Bruce et al., 2024) scales this framework to large transformer-based architectures, focusing on latent-action-driven world model prediction in addition to policy learning. A few works (Ye et al., 2025; NVIDIA et al., 2025; Bu et al., 2025; Chen et al., 2025) have also explored the utility of latent action learning in embodied agents, particularly in the vision–language–action setting. Our work differs from prior approaches in that we leverage a pre-trained video generation model to co-evolve latent action learning and world modeling, a direction that has not been explored before.

**Latent-action-based World Models** While the FDM in the latent action model can be interpreted as a world model, most works do not explicitly focus on future prediction abilities, with the exception of (Cui & Gao, 2023). However, the prediction quality of FDMs is generally lower than that of high-capacity video-generation-based world models. Recently, Genie (Bruce et al., 2024) trained a separate decoder-only MaskGIT (Chang et al., 2022) as the world model, conditioned on a fixed latent action space learned beforehand. AdaWorld (Gao et al., 2025) is the work most closely related to ours, adopting a similar two-stage approach as Genie but using a diffusion-based video model and extending discrete latent actions to continuous ones. Other efforts, such as AD3 (Wang et al., 2024b) and PreLAR (Zhang et al., 2024), integrate latent action learning with dynamics and policy training in a Dreamer-style (Hafner et al., 2021) architecture trained from scratch, rather than leveraging the benefits of large-scale pre-trained video generation models.

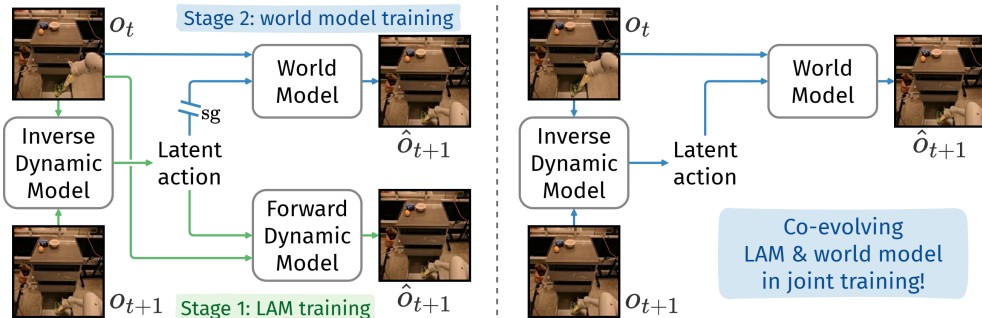

Figure 1: **(a)** Prior works use a two-stage pipeline: learn a latent action model (LAM), then fix it to train the world model. **(b)** We propose a one-stage pipeline, directly using the world model as the forward dynamics model and backpropagating gradients through latent actions.

**Finetuning Pre-trained Video Generation Model as World Models** Our work is also related to efforts that fine-tune pre-trained video generation models into controllable world models by adding action conditioning. Except for AdaWorld (Gao et al., 2025) discussed above, most works in this line assume a pre-specified action space. AVID (Rigter et al., 2025) introduces a lightweight adapter on top of a frozen video generation model for action conditioning and world modeling. IRASim (Zhu et al., 2024) uses adaptive layer normalization (Peebles & Xie, 2023) to incorporate actions, analogous to how text prompting is conditioned. Following IRASim, DWS (He et al., 2025) proposes a more granular action conditioning mechanism along with other improvements for world modeling. Vid2World (Huang et al., 2025a) focuses on challenges of temporal causality in adapting video diffusion models to world models, while EnerVerse-AC (Jiang et al., 2025) adds action conditioning to a robotics foundation model (Huang et al., 2025b) for manipulation tasks.

## 3 METHOD

### 3.1 WORLD MODELS WITH LATENT ACTIONS

We focus on training a world model to predict the next observation $o_{t+1}$ based on the current observation $o_t$ and a *latent action* $z_t$, modeling the distribution $p(o_{t+1} \mid o_t, z_t)$. Unlike pre-specified actions, such as keyboard or mouse inputs in video games, latent actions are learned entirely from observational data. This allows us to pre-train world models on large-scale, action-free video data.

As mentioned in the introduction, previous works (Bruce et al., 2024; Gao et al., 2025) typically adopt a two-stage process, training a latent action model (LAM) prior to world model training. The LAM consists of an inverse dynamics model (IDM) and a forward dynamics model (FDM). Specifically, the IDM $f_{\text{inv}}$ takes the current observation $o_t$ and the next observation $o_{t+1}$ as input and outputs a latent action $z_t$, while the FDM $f_{\text{fwd}}$ takes $o_t$ and $z_t$ to predict the next observation $\hat{o}_{t+1}$. LAM is trained by minimizing the reconstruction loss between $\hat{o}_{t+1}$ and $o_{t+1}$, *i.e.*,

$$\mathcal{L}_{\text{LAM}} = \|o_{t+1} - f_{\text{fwd}}(o_t, f_{\text{inv}}(o_t, o_{t+1}))\|. \tag{1}$$

To prevent trivial solutions, a bottleneck is often applied to the latent action space, forcing the latent actions to compactly encode the most meaningful changes between $o_t$ and $o_{t+1}$. Once trained, the IDM is frozen and used to extract latent action labels for observation sequences. Previous works then train a separate world model to capture $p(o_{t+1} \mid o_t, z_t)$, typically employing a much higher-capacity model than the LAM. The complete pipeline is illustrated in Figure 1(a).

However, one may immediately notice that the FDM and the world model perform exactly the same task: predicting $o_{t+1}$ based on $o_t$ and $z_t$. Our idea is to replace the FDM with the world model, reducing the two-stage training into a single joint training framework that performs dynamics learning and latent action learning simultaneously in an end-to-end fashion, as illustrated in Figure 1(b). Such a framework not only enables a more elegant model design and efficient training but also allows the co-evolution of latent actions and the world model. The powerful world model can provide gradients that help the IDM learn higher-quality latent actions, while the IDM produces a more informative latent action space, offering the world model a clearer control interface.

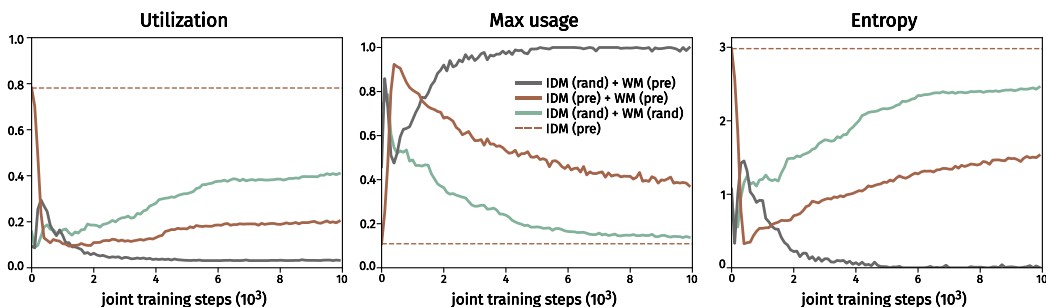

Figure 2: Latent action codebook metrics during joint training of the IDM and world model. "rand" indicates random initialization, while "pre" indicates initialization from pre-trained weights. The dashed line shows the codebook metrics of the pre-trained IDM. All three subplots share the same legend, shown only in the middle panel for clarity.

While this idea may seem simple, we show in the next subsection that naively training the inverse dynamics model and the world model together can easily collapse. One might also argue that the FDM is essentially a world model and could be used to roll out future predictions. Empirically, however, we find that the FDM produces much lower-quality predictions than a separately trained world model. We believe this explains why previous works adopt a two-stage approach. To the best of our knowledge, no prior work has successfully attempted this type of joint training.

## 3.2 TAMING THE FRAGILITY OF JOINT TRAINING

Following prior work (Bruce et al., 2024; Gao et al., 2025), we instantiate the IDM in Figure 1(b) as an ST-Transformer (Xu et al., 2020), followed by vector quantization (Van Den Oord et al., 2017) to produce discrete latent actions. For the world model, we adopt OpenSora (Zheng et al., 2024), a high-performing open-source diffusion-based video generative model. We choose OpenSora for its demonstrated effectiveness in the DWS method (He et al., 2025), where it was adapted for world modeling with pre-specified actions. Additional implementation details are deferred to Section 3.3.

When training the model, however, we observe that learning quickly collapses. As shown by the gray curve in Figure 2, the utilization rate of the VQ codebook for the latent actions drops to zero after an initial brief increase. At the same time, the maximum code usage rapidly rises to nearly 100%, indicating that the model collapses to using only a very small subset of latent actions. The concurrent drop of code entropy to zero further suggests that all codes in the codebook degenerate into a single dominant code. In contrast, a healthy latent action codebook should exhibit relatively high utilization and entropy, along with low maximum usage, as indicated by the dashed horizontal lines in Figure 2.

As we have seen, directly training a freshly initialized IDM jointly with a pre-trained world model leads to collapse. We hypothesize that this occurs because the powerful, pre-trained world model quickly learns to disregard the random and uninformative action signals provided by the from-scratch LAM. By relying on its own strong internal priors to minimize the prediction loss, the world model provides no structured, supervisory gradient back to the LAM, causing its representation to degenerate into a few dominant, uninformative codes. To further investigate the fragility of joint training, we next initialize the IDM using parameters from a reasonably well-trained latent action model (corresponding to the dashed horizontal lines in Figure 2). However, as the brown curve in Figure 2 shows, even though it starts from a favorable state, the codebook quickly deteriorates, leading to low utilization and entropy. Although it gradually improves later, the progress remains too slow to be practical.

Given that neither random nor guided initialization works, we hypothesize that the IDM is not well aligned with the pre-trained weights of the world model. To test this, we randomly initialized both the IDM and the world model and trained them jointly. As shown by the green curve in Figure 2, this setup does not collapse, supporting our hypothesis. To mitigate the instability while still taking advantage of powerful pre-trained video generation models, we propose a warm-up strategy: first train the IDM while keeping the world model frozen, then switch to joint training.

With this warm-up, the IDM is able to catch up with the world model, enabling stable joint training without collapse. As the dark blue curve in Figure 3 shows, the codebook metrics remain healthy

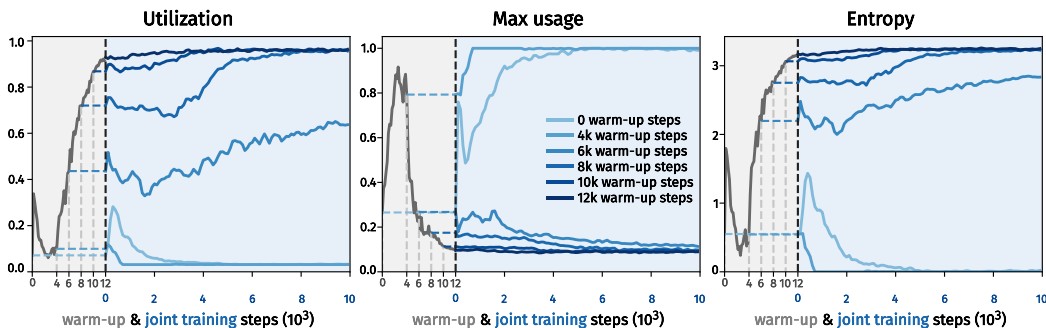

Figure 3: Latent action codebook metrics during warm-up and joint training. Different blue curves correspond to IDM initializations from warm-up checkpoints at various steps. All three subplots share the same legend, shown only in the middle panel for clarity.

under this scheme. We further varied the number of warm-up steps. Figure 3 shows that longer warm-up generally leads to more stable subsequent joint training, confirming that the IDM indeed undergoes a catch-up phase during warm-up. In practice, we choose a warm-up length that ensures stability while reserving as many steps as possible for end-to-end co-evolution.

After warm-up, we jointly train the IDM and world model end-to-end, allowing them to co-evolve and adapt to each other. The world model provides gradients that guide the IDM to learn higher-quality latent actions, while the IDM in turn produces a more informative latent action space for the world model. In Section 4, we present extensive experiments showing that this joint training strategy enhances both the quality of the learned latent actions and the performance of the world model.

### 3.3 IMPLEMENTATION DETAILS

We elaborate on the key implementation details central to our joint training paradigm, focusing on the latent action conditioning mechanism and the end-to-end training process. Further information regarding model architectures and training details are deferred to the Appendix B.

**Latent Action Conditioning**. We integrate latent actions extracted by the IDM into the pre-trained OpenSora model via Adaptive Layer Normalization (AdaLN) (Peebles & Xie, 2023). The sequence of the latent actions is first processed by a from-scratch self-attention network to produce contextualized embeddings. These embeddings are then projected into action-specific scale, shift and gate parameters by a MLP, which are then fused via addition with the original modulation parameters derived from the diffusion timesteps, and applied at each LayerNorm layer within all the OpenSora blocks. This mechanism provides control signals to condition the denoising process on the latent actions.

**Training Objective and Gradient Flow.** The system is jointly optimized using a flow matching loss objective (Liu et al., 2022a) provided by the OpenSora model, which learns to predict the velocity needed to denoise the video latent. The warm-up and end-to-end training phases carefully manage the gradient flow generated by the loss. During warm-up, the pre-trained OpenSora model is frozen, and the loss is backpropagated through the action AdaLN parameters and solely update the action conditioning modules and the LAM components (IDM and VQ quantizer). Subsequently, in the end-to-end phase, we unfreeze the OpenSora world model and the unified gradient updates all components simultaneously. Crucially, this end-to-end gradient flow is the core mechanism for synergistic co-evolution.

## 4 EXPERIMENTS

In this section, we conduct experiments to answer the following questions:

1. How does our joint training paradigm compare against the traditional two-stage approach in terms of LAM representation quality and world model video prediction performance?

2. What is the underlying mechanism of our paradigm's success? Do the LAM and the World Model truly engage in a synergistic co-evolution during joint learning?

3. Can the inherent advantages of our joint training paradigm translate into performance gains in practical real-action-based video simulation?

4. What is the ultimate efficacy of CoLA-World as a learned simulator for solving control tasks via visual planning?

## 4.1 EXPERIMENTAL SETUP

**Dataset** We focus on learning latent-action-based world models for robotic manipulation that can adapt to diverse downstream embodiments and action spaces. Our training data consists of a large-scale mixture of human egocentric videos and robot manipulation videos. Importantly, the training process is entirely action-free: both the world model and the latent action model are learned purely from video. Full dataset details are provided in Appendix A.

**Baselines** We compare two training paradigms. **2-STAGE**: Following prior work, we first train a LAM (comprising an IDM, an FDM, and a VQ quantizer) from scratch. Then the LAM is frozen and its IDM and quantizer are used to provide latent actions for fine-tuning the world model, while the FDM is discarded. **JOINT** (CoLA-World): Our joint learning paradigm begins with a brief warm-up phase to align the from-scratch LAM (IDM and quantizer) with the pre-trained world model, followed by full end-to-end (E2E) joint training. The architectures of the LAM and world model are identical across both paradigms. In the 2-stage setting, we train the LAM for 30K steps to ensure a high-quality representation. For joint training, we use an 8K warm-up phase (Figure 3), which provides a stable initialization while preserving budget for the E2E phase. Additional training details are provided in Appendix B. For clarity, we denote checkpoints by training budgets of their respective phases, *e.g.*, LAM30K + WM30K in 2-stage learning; WARM8K + E2E52K in joint learning.

**Evaluation metrics**. To assess the quality of the learned latent action, we employ a linear probing task, where a simple one-layer linear projection head is trained to predict the original real action from the frozen latent actions. Here we evaluate on L1 prediction loss to prevent potential outliers dominating the loss results. For the world model, we measure action-conditioned video generation quality using a suite of standard metrics: PSNR, SSIM, LPIPS and FVD. In the tables, LPIPS and SSIM scores are scaled ×100 for compact display.

## 4.2 PERFORMANCE OF THE JOINTLY LEARNED LAM AND WORLD MODEL

Table 1: Linear probing loss across several robotics datasets (lower is better).

| METHOD | | BRIDGE | RT-1 | KUKA | DROID | AGIBOT | LIBERO |
|---|---|---|---|---|---|---|---|
| 2-STAGE | LAM30K | 0.0827 | **0.1191** | 0.0741 | 0.1912 | 0.1035 | **0.1614** |
| JOINT | WARM8K + E2E22K | **0.0815** | 0.1206 | **0.0736** | **0.1911** | **0.0908** | 0.1623 |

**Latent Action Quality.** We first evaluate the quality of the learned latent action representations via linear probing on six robotics datasets, including five from the Open X-Embodiment suite (Collaboration et al., 2023) and one out-of-distribution LIBERO dataset (Liu et al., 2023) unseen during training. As shown in Table 1, our CoLA-World yields a competitive latent action space, achieving lower probing loss on most datasets.

While the difference in probing loss appears marginal, this isolated metric does not fully capture the latent action representation's utility. The ultimate measure of a latent action's quality lies in its ability to effectively control the world model. As we will show, the world model guided by the jointly learned LAM significantly outperforms the two-stage baseline on LIBERO. This suggests that our co-evolved latent action space, while less amenable to linear probing, provides a more robust and effective control interface for world modeling.

**World Model Simulation Performance.** We then evaluate the latent-action-conditioned video prediction performance of the world model. Table 2 reports results across several in-distribution datasets (OXE, EgoCentric, AgiBot) and one out-of-distribution (LIBERO) dataset, comparing different training checkpoints. With the same total training budget of 60K steps, our joint training paradigm (WARM8K + E2E52K) consistently matches or surpasses the best two-stage method (LAM30K + WM30K) across all datasets. Notably, improvements are most pronounced on the

Table 2: Video prediction performance of the learned world models on different datasets.

| DATASET | | METHOD | PSNR ↑ | SSIM ↑ | LPIPS ↓ | FVD ↓ |
|---|---|---|---|---|---|---|
| OXE | 2-STAGE | LAM30K + WM30K | 22.34 | 81.16 | 13.17 | 291.30 |
| | | LAM8K + WM52K | 21.91 | 80.76 | 13.79 | 296.64 |
| | JOINT | WARM8K + E2E52K | **22.57** | **81.40** | **12.79** | **278.90** |
| | | WARM8K + E2E30K | 22.26 | 81.06 | 13.26 | 289.37 |
| EGOCENTRIC | 2-STAGE | LAM30K + WM30K | **23.80** | **83.68** | **12.90** | 260.14 |
| | | LAM8K + WM52K | 23.48 | 83.28 | 13.46 | 267.94 |
| | JOINT | WARM8K + E2E52K | 23.69 | 83.52 | 13.08 | **252.45** |
| | | WARM8K + E2E30K | 23.66 | 83.41 | 13.26 | 263.57 |
| AGIBOT | 2-STAGE | LAM30K + WM30K | 23.61 | 85.36 | 10.11 | 185.63 |
| | | LAM8K + WM52K | 23.30 | 85.11 | 10.30 | 196.18 |
| | JOINT | WARM8K + E2E52K | **23.93** | **85.61** | **9.86** | **174.93** |
| | | WARM8K + E2E30K | 23.64 | 85.27 | 10.22 | 189.03 |
| LIBERO | 2-STAGE | LAM30K + WM30K | 23.13 | 86.90 | 10.22 | 167.77 |
| | | LAM8K + WM52K | 22.72 | 86.43 | 10.78 | 190.09 |
| | JOINT | WARM8K + E2E52K | **23.33** | **87.21** | **9.89** | **158.36** |
| | | WARM8K + E2E30K | 23.25 | 87.05 | 10.08 | 164.86 |

perceptually aligned FVD metric, indicating that our generated videos are not only pixel-accurate but also more temporally coherent and realistic.

Crucially, our paradigm also demonstrates superior sample efficiency. Our WARM8K + E2E30K model, with a substantially smaller budget, already approaches the performance of the fully trained LAM30K + WM30K 2-stage model and surpasses it on the out-of-distribution LIBERO dataset. This efficiency arises from the synergistic training, which avoids the redundant learning and static bottlenecks inherent in the 2-stage approach. Moreover, when the 2-stage method is given a similar total budget (LAM8K + WM52K *vs.* WARM8K + E2E52K), it is significantly outperformed, even lagging behind our less-trained WARM8K + E2E30K checkpoint due to its under-trained, static LAM. These results highlight that our joint training unlocks a higher performance ceiling with significantly fewer training steps. We provide latent action transfer results in Appendix D.2.

### 4.3 EVIDENCE FOR SYNERGISTIC CO-EVOLUTION

Having shown the performance of our CoLA-World, we now turn to understanding the mechanism behind its success. To this end, we design two controlled ablation studies to dissect the bidirectional information flow and verify the presence of a virtuous cycle of mutual promotion.

**An Evolving World Model as a Better Tutor for the LAM.** To isolate the influence of the world model's own learning process on the LAM, we compare our WARMUP + E2E method with a PURE WARMUP variant, where the LAM is trained using gradients from a frozen world model. We evaluate the resulting LAMs via linear probing loss on the LIBERO dataset, as shown in Figure 4(a). While the LAM guided by the static world model (PURE WARMUP) improves steadily, the LAM in our CoLA-World exhibits much faster reduction in probing loss once E2E training starts. This demonstrates that the supervisory signal from the world model evolves over time: as the world model refines its own understanding of the world's dynamics, the gradients it provides to the LAM become progressively more informative and causally sound. These results confirm that a concurrently improving world model acts as a effective tutor, enabling a better and more efficiently learned LAM.

**An Evolving LAM as a Better Control Interface for the World Model.** We then investigate the impact of a dynamically evolving LAM on the world model's video prediction performance. We compare our WARMUP + E2E model against a variant where the LAM is frozen after the same initial warmup phase and only the world model is fine-tuned subsequently. As shown in Figure 4(b), the world model paired with a frozen LAM improves initially but quickly plateaus. By contrast, when paired with a continuously improving LAM during E2E training, the world model achieves substantially higher video generation quality. This demonstrates that a static latent action space

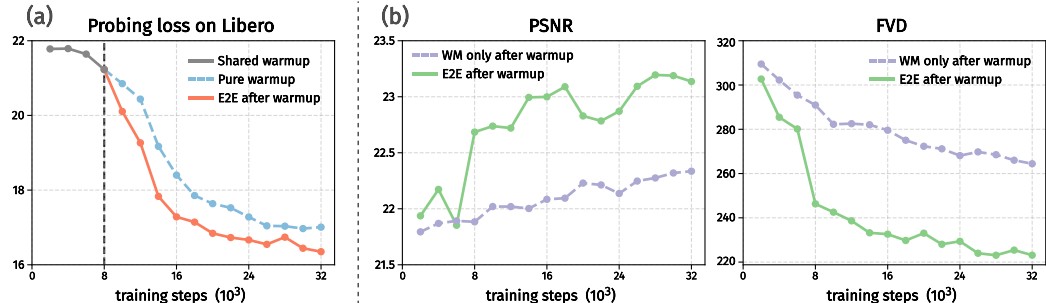

Figure 4: Evidence of synergistic co-evolution. The LAM's probing loss drops faster when the world model is co-evolving (**a**), while the world model achieves higher video prediction performance as the LAM improves (**b**).

imposes a performance bottleneck, whereas a dynamically evolving LAM provides a progressively more precise control interface, unlocking the world model's full predictive potential.

**The Virtuous Cycle of Co-evolution.** These two experiments provide evidence for a virtuous cycle of synergistic co-evolution: an improving world model better shapes the latent action representation, which in turn enables more effective world modeling. This dynamic co-evolution creates a deeply coupled and intrinsically consistent system. As shown in the following section, this property underlies our model's superior performance on downstream adaptation tasks.

## 4.4 ADAPTATION FOR REAL-ACTION-BASED SIMULATION

A key promise of latent-action-based world models is their adaptability to diverse, real-action control interfaces. We evaluate this capability by adapting our world model to new, out-of-distribution robotic environments including LIBERO and RoboDesk (Kannan et al., 2021).

**Adaptation and Evaluation Protocol** For each downstream dataset, we follow Gao et al. (2025) and first train a lightweight two-layer MLP adapter to map the dataset's real actions to the latent actions. Subsequently, we fine-tune the world models for 3K steps. Crucially, this fine-tuning is performed using ground-truth latent actions (GT-LAM), which are extracted from the downstream videos by the frozen learned LAM. This ensures the world model learns the new environment's dynamics from the clean supervisory signal, consistent with its pre-training. Finally, we evaluate the fine-tuned world model in two distinct modes: (a) using the same GT-LAM to assess the ideal performance ceiling after domain-specific finetuning, and (b) using the trained adapter to translate real actions into latent actions and assess the world model's practical, real-action-based video prediction performance.

**Results and Analysis**. To evaluate our paradigm's efficiency, we compare our jointly trained WARM8K + E2E30K checkpoint against the more extensively trained LAM30K + WM30K two-stage model. Despite using a smaller training budget, Table 3 shows that CoLA-World clearly outperforms the two-stage baseline. In GT-LAM evaluation, it already demonstrates an advantage, indicating that the jointly trained world model provides a stronger foundation for learning dynamics in unseen environments.

Moreover, the performance gap between CoLA-World and the two-stage baseline becomes more pronounced when evaluated with real actions, particularly on the FVD metric. This reflects a fundamental distinction in how the LAM and world model interact under the two paradigms. The two-stage model, fine-tuned on a fixed GT-LAM distribution, becomes rigidly calibrated to this static representation. When faced with biased latent actions from an imperfect adapter, the world model struggles to interpret these out-of-distribution signals, leading to a substantial performance drop.

By contrast, our world model co-evolves with a dynamically improving LAM, continually adapting to a smoothly changing latent action landscape. This process endows the world model with a more smooth and robust utilization of the latent action space, making it more resilient to the adapter's imperfections, correctly interpreting its biased outputs as functionally equivalent to the ground-truth signals. This intrinsic consistency allows CoLA-World to generalize effectively from ideal training

Table 3: Video prediction performance of the finetuned world models, taking latent actions inferred by the LAM or translated from the real actions by the learned adapters as conditions.

| Dataset | Action Type | Method | PSNR ↑ | SSIM ↑ | LPIPS ↓ | FVD ↓ |
|---------|-------------|--------|--------|--------|---------|-------|
| LIBERO | GT-LAM | LAM30K + WM30K | 25.51 | 89.55 | 7.41 | **73.54** |
| | | WARM8K + E2E30K | **25.85** | **89.82** | **7.31** | 74.65 |
| | REAL ACTION | LAM30K + WM30K | 22.45 | 86.96 | 9.56 | 115.45 |
| | | WARM8K + E2E30K | **22.68** | **87.15** | **9.27** | **93.68** |
| ROBODESK | GT-LAM | LAM30K + WM30K | 24.21 | 86.99 | **7.41** | 120.51 |
| | | WARM8K + E2E30K | **24.29** | **87.04** | 7.57 | **120.26** |
| | REAL ACTION | LAM30K + WM30K | 20.03 | 83.33 | 10.64 | 188.82 |
| | | WARM8K + E2E30K | **21.37** | **84.67** | **8.90** | **169.70** |

Table 4: Visual planning success rate on RoboDesk in the VP$^2$ benchmark.

| Method | Upright Block | Push Slide | Flat Block | Push Drawer | Average |
|--------|---------------|------------|------------|-------------|---------|
| 2-STAGE | 20.0% | 4.44% | 1.11% | 2.22% | 6.94% |
| JOINT | 37.78% | 6.11% | 3.33% | 5.25% | 13.12% |

signals to practical real-world control interfaces. Furthermore, as shown by a quantitative analysis of codebook metrics in Appendix D.1, the latent action space learned through joint training proves robust to the potential representation collapse observed in the two-stage approach during downstream adaptation, preserving its diversity and thus validating its strong generalization performance.

## 4.5 Visual Planning

To evaluate the final utility of our world model for downstream control, we assess the planning performance of our adapted world models using the VP$^2$ benchmark (Tian et al., 2023). We take the CoLA-World and two-stage models previously fine-tuned on the RoboDesk dataset and evaluate their ability to solve four challenging manipulation tasks using a sampling-based Model Predictive Control planner. The results, summarized in Table 4, indicate that our CoLA-World paradigm demonstrates a clear advantage over the two-stage approach, especially on Upright Block task. This confirms that the superior simulation quality demonstrated in Section 4.4 translates into more reliable prediction results for the planner, leading to more effective control.

On several complex tasks, both methods exhibited low performance, underscoring the inherent difficulty of these high-precision manipulation problems for any planner relying purely on a learned visual model. Nevertheless, the consistent and sometimes substantial performance gains achieved by CoLA-World on the tractable tasks strongly validate our joint training methodology as a more effective foundation for real-world control applications.

## 5 Conclusion, Limitation and Future Work

In this work, we introduce CoLA-World, the first framework to successfully realize the synergistic joint training of a latent action model with a pre-trained video-generation-based world model. A critical warmup phase resolves the inherent instability of this approach, enabling co-evolution between latent action learning and world modeling. Our experiments show that CoLA-World significantly outperforms previous two-stage methods in both simulation quality and downstream planning. A potential limitation is that the world model's performance depends on the pre-trained video generation model and requires substantial computational resources; however, this can be mitigated with more efficient models, and our paradigm is broadly applicable for injecting latent action conditioning. Future directions include evaluating the learned latent actions in vision-language-latent-action settings (Chen et al., 2025; Bu et al., 2025) for robotic manipulation policy training, and scaling our framework to train foundational world models on larger video datasets for broader adaptability.

## REPRODUCIBILITY STATEMENT

We are committed to ensuring the reproducibility of our results. All datasets used in our experiments are publicly available, with detailed descriptions provided in Appendix A. Comprehensive information on model architectures and training protocols can be found in Appendix B. Our code is available in an anonymous repository for review at `https://anonymous.4open.science/r/CoLA-World`, and model checkpoints will be released upon publication.

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

## LARGE LANGUAGE MODELS (LLMS) USAGE

We used large language models (LLMs) solely as a writing and code-assistance tool, for tasks such as polishing text and providing autocomplete suggestions in code. The LLMs did not contribute to the research ideation, experimental design, data analysis, or interpretation of results. All scientific content, results, and conclusions are the original work of the authors.

## A  DATASET

We mainly focus on learning a latent action model and a world model for robotic manipulation that is adaptable to diverse downstream embodiments and action spaces. The data mixture for CoLA-World training is composed of both robot videos and human manipulation videos. For robot data, we primarily use Open X-Embodiment (OXE) (Collaboration et al., 2023) mixture and the AgiBot (AgiBot-World-Contributors et al., 2025) dataset. For human videos, we curate a comprehensive collection from nine prominent datasets, including Something-Something V2 (Goyal et al., 2017), RH20T (Fang et al., 2023), Ego4D (Grauman et al., 2022), EgoPAT3D (Li et al., 2022), EGTEA Gaze+ (Li et al., 2018), HOI4D (Liu et al., 2022b), EPIC-KITCHENS (Damen et al., 2020), HO-Cap (Wang et al., 2024a) and HoloAssist (Wang et al., 2023). The final data mixture consists of approximately 30% OXE, 20% AgiBot, and 50% human video data.

## B  IMPLEMENTATION DETAILS

Our two-stage training baseline involves training a LAM consisting of an IDM and an FDM, as well as a VQ quantizer to bottleneck the latent action space. Then the latent actions are inferred from the video using the frozen IDM and quantizer, used to finetune a pre-trained OpenSora video generation model into a world model, while the FDM is discarded. The joint training paradigm trains the LAM (i.e. the IDM and the VQ quantizer) and the OpenSora world model simultaneously, detaching the gradients of the world model's weights when executing warm-up. For fair comparison, the architectures of the IDM, the quantizer and the world model as well as the action conditioning moodules of the two paradigms are totally the same. We then elaborate each of the mentioned components above.

### B.1  IDM, FDM AND THE QUANTIZER

The IDM is implemented as an 12-layer ST-Transformer (Xu et al., 2020). Each block has a hidden dimension of 768 and 12 attention heads. The FDM is implemented as an 12-layer spatial Transformer with the same number of hidden dimension and attention heads as the IDM. Between the IDM and FDM, we apply vector quantization (Van Den Oord et al., 2017) to produce latent actions, which is composed of two 32-dimensional action tokens chosen from the codebook. The codebook contains 32 entries, yielding a total number of 1024 different latent action choices. The IDM takes an $T \times 224 \times 224 \times 3$ video clip as input, first patchified with a patch size of 14 and then processed by the ST-Transformer to predict $T - 1$ latent actions. The FDM concatenates the image patches and the predicted latent action tokens, using the spatial transformer to produce pixel decoding results of the next frames. The IDM and FDM both have about 0.12 B parameters.

### B.2  WORLD MODEL BASED ON THE PRETRINED OPENSORA MODEL

We adopt the pre-trained OpenSora model as the backbone of the world model. We use the v1.2 release with about 1.2 B parameters. As mentioned in Section 3.3, we add an extra from-scratch module for conditioning the video generation of OpenSora on the extracted latent actions, including 6 self-attention blocks to process the latent action sequence and an MLP to get the final AdaLN parameters of the latent actions, which are then fused with original diffusion timestep AdaLN parameters and modulate the attention results in each OpenSora DiT block. We initialize the weights in the action attention blocks as zero, to ensure a steady training at the beginning. Similar AdaLN-style action conditioning method is also explored in previous work  (Zhu et al., 2024; He et al., 2025). However, their action inputs are fixed and not learnable, while our latent actions and conditioning layers are dynamically refined by the world model's own objective, which sets our method apart.

These newly introduced from-scratch modules to the OpenSora have about 74M parameters. The original layers in OpenSora for processing the texts, as well as the cross attention layers for fusing visual and text modalities, are discarded. Then there the about 0.93 B learnable parameters in the OpenSora, including the newly added action conditioning modules. Moreover, the original temporal transformer blocks in the OpenSora Dit are not causal, and we add causal masks in them to prevent future information influencing the past, which is unfavorable in dynamics modeling.

During training, the OpenSora WM takes in 256-resolution videos and the extracted latent action sequence, adding noise to the ground-truth videos and forwarding them through the DiT to predict the velocity vector, and building the prediction loss in the context of rectified flow. We use a timestep-wise classifier-free guidance, where during training we randomly mask the action condition as zero in a probability of 0.1 at each timestep of the sequence, and apply a guidance scale of 4.0 for sampling during inference. The number of denoising timestep is 10 in inference.

### B.3 TRAINING DEATILS

**Latent Action Training of the two-stage paradigm** After FDM producing pixel reconstruction results, we simply build the MSE loss between the reconstruction and the ground-truth "next frame" observation, in a teacher-forcing manner, rather than multi-step auto-regression. The vq loss and commitment loss introduced by the vq technique are also included to update the IDM and the codebook, and their loss weights are 1.0.

**World Model Training of the two-stage paradigm** As mentioned above, the OpenSora world model builds the flow matching loss using the input videos and the detached latent actions and update the OpenSora model, as well as the action conditioning modules.

**Training of CoLA-World paradigm** The OpenSora world model now builds the flow matching loss using the input videos and the learnable latent actions. The gradients then backpropagate throughout the whole system. The IDM, VQ quantizer and the action conditioning modules introduced in the OpenSora will be updated, while the pretrained weights of the original OpenSora model will only be updated after warm-up phase. The IDM and VQ quantizer will also receive gradients from the vq loss and commitment loss both during warm-up and end-to-end phase, similar to the latent action training in the two-stage paradigm.

**Other training protocols.** To ensure fair comparison, both training paradigms use a learning rate of 7.5e-5, a batch size of 128, and a 2000-step linear warmup schedule for the learning rate. When the LAM model is updating (LAM training of 2-stage paradigm, and all of the joint training paradigm), we use random crop to the video clips as a data augmentation trick to improve performance, while when the LAM is fixed, we do not use the augmentation and direct use the IDM to extract the latent actions from the original video.

## C   EVALUATION DETAILS

### C.1   EVALUATION SETUP

For linear probing task and all the video prediction tasks, we train the prober head (the LAM and the world model) on the training split of the given dataset mixture, and validate on the valid split. For example, for linear probing on out-of-distribution LIBERO dataset, in fact the LAM is previously trained on the whole training data, and the prober head is now trained on the training split of the unseen LIBERO dataset. Then, we test the performance of the LAM and the prober by probing the loss on the valid split of the LIBERO dataset, and record the results. For all the probing tasks, we train the prober head for 1K steps with a batch size 64 on 8 gpus (512K samples in all), and validate on 20K test samples. For all the video prediction tasks, we evaluate on a fixed test dataset for each data mixture, consisting of 240 video clips on each gpu, and the performance is averaged.

### C.2   REAL ACTION ADAPTATION

When adapting the trained world model to a downstream real action space, we first train the adapter predicting the GT-LAM vq code indices from the real actions using a 2-layer MLP. This takes 1K

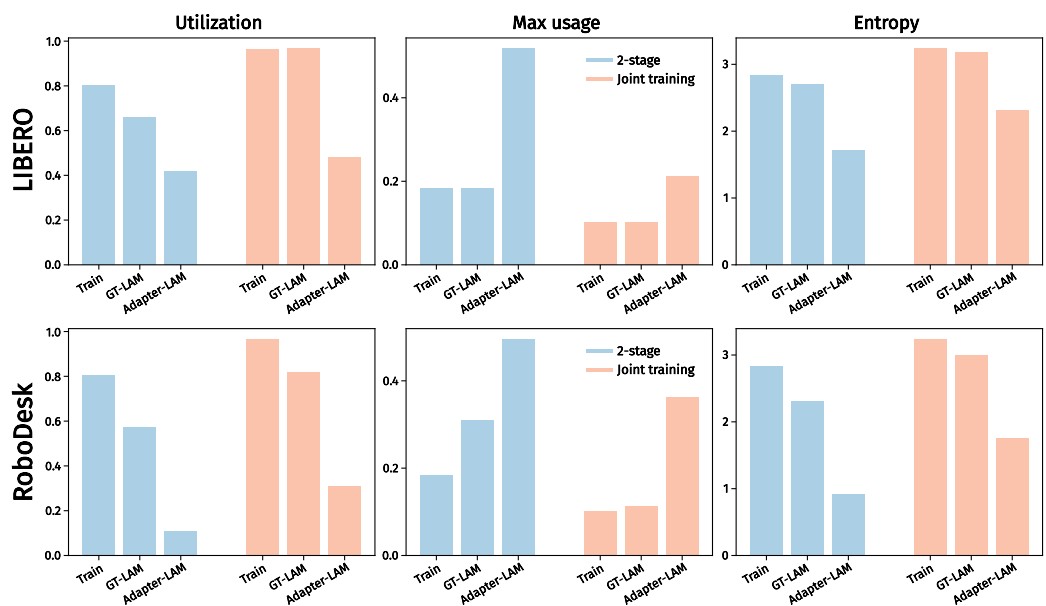

Figure 5: Codebook metrics in different training and adaptation stages. All subplots share the same legend, shown only in the middle panel for clarity.

steps training with a batch size of 64. We then finetune the world model on downstream dataset using Gt-LAM for 3K steps with a batch size of 16.

### C.3 VISUAL PLANNING ON VP$^2$ ENCHMARK

We test the learned world model's utility in control on RoboDesk environment using the evaluation protocol from VP$^2$ benchmark. Each task of the RoboDesk environment in VP$^2$ benchmark is specified by 30 pairs of initial observation and goal observation. When testing on one task, every time we sample such a pair and the agent needs to use the world model to plan the trajectory starting at the initial state towards the goal. The reward function is also provided by VP$^2$, defined as the weighted sum of the MSE loss between the predicted video and the goal observation, with a pretrained binary classifier's predicted logit on the current task. The classifier's weights are also provided by the benchmark. Finally, the task success rate is the ratio of success trajectories in these 30 runs. Moreover, VP$^2$ offers trajectory data on RoboDesk, and the experiments of world model downstream adaptation on RoboDesk in Section 4.4 is conducted by training the adapter and finetuning the world model on these data.

## D ADDITIONAL RESULTS

### D.1 ANALYSIS OF CODEBOOK DYNAMICS IN DOWNSTREAM ADAPTATION

To provide deeper quantitative insight into the mechanisms behind our paradigm's superior downstream real-action-adaptation performance over two-stage method, we analyze the metrics of the VQ codebook. For both CoLA-World and the Two-Stage baseline, we compare three distinct latent action distributions on the LIBERO and RoboDesk datasets:

(a) Training Distribution: The latent action distribution in our general training.

(b) GT-LAM Fine-tuning Distribution: The ground-truth latent action distribution inferred by the frozen LAM encoder from the downstream task videos, used for fine-tuning the world model.

(c) Adapter-LAM Inference Distribution: The latent action distribution produced by the trained adapter when translating the downstream task's real actions.

The results, visualized in Figure 5, reveal a stark contrast in how the two paradigms adapt their latent action space.

As shown in the bar charts, the two-stage method exhibits a dramatic representational collapse when adapting to the downstream tasks' real actions. While the codebook utilization and entropy are reasonable during pre-training (a), they decrease when the model is fine-tuned on the narrower distribution of the downstream GT-LAM (b). Most critically, when the adapter is used for inference (c), the codebook metrics degenerate severely and tend to collapse: codebook utilization plummets to nearly 10% on RoboDesk, with the max_usage metric spiking to approximately 0.5 on both LIBERO and RoboDesk. This indicates that the adapter has found a "lazy shortcut" by mapping the vast majority of real actions to a single, all-purpose latent code. This is a direct cause of the model's low performance and its inability to handle the full complexity of the control task.

In contrast, the overall codebook usage is relatively healthy in our CoLA-World paradigm under the Adapter-LAM setting. The entropy remains high and the max_usage stays at a relatively low level compared to the two-stage baseline. This provides direct, quantitative evidence that the co-evolutionary process has forged a more robust and flexible latent action space for downstream adaptation and generalization. The constant, supervisory feedback from the powerful world model tutor prevents the LAM from taking degenerative shortcuts, compelling them to learn a richer, more meaningful representations. This preserved diversity of the codebook is a cornerstone of our system's adaptation performance and its ability to robustly generalize.

To conclude, and in conjunction with our analysis in Section 4.4, our joint training paradigm's success in downstream adaptation stems from co-evolution forging an intrinsically consistent and deeply coupled system, which manifests in the dual advantages of a collapse-resistant latent action space and a world model that robustly utilizes it.

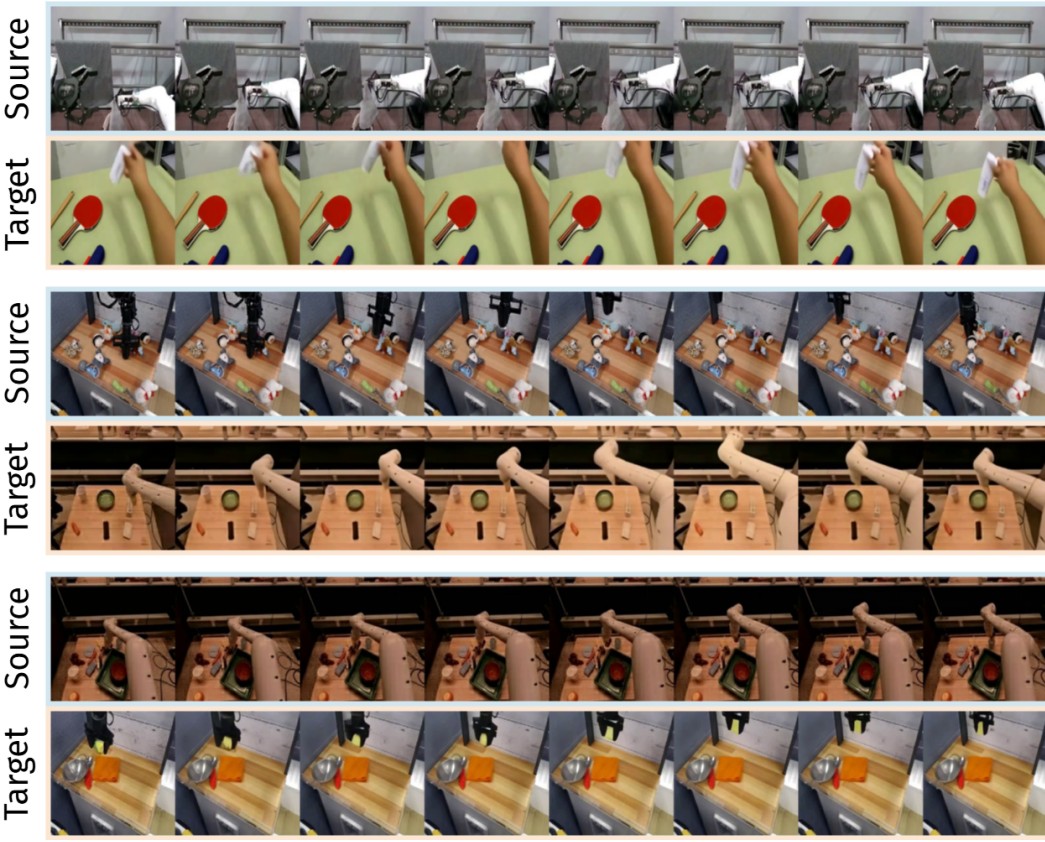

Figure 6: Action transfer results. The first image of the generated video comes from a different dataset from the source video.

Table 5: Video prediction performance of the learned world models on different datasets.

| DATASET | METHOD | | PSNR ↑ | SSIM ↑ | LPIPS ↓ | FVD ↓ |
|---|---|---|---|---|---|---|
| OXE | 2-STAGE | LAM30K + WM52K | 22.54 | **81.45** | 12.87 | 281.05 |
| | JOINT | WARM8K + E2E52K | **22.57** | 81.40 | **12.79** | **278.90** |
| EGOCENTRIC | 2-STAGE | LAM30K + WM52K | **23.72** | **83.56** | **12.96** | 259.33 |
| | JOINT | WARM8K + E2E52K | 23.69 | 83.52 | 13.08 | **252.45** |
| AGIBOT | 2-STAGE | LAM30K + WM52K | 23.79 | 85.57 | 9.91 | 180.45 |
| | JOINT | WARM8K + E2E52K | **23.93** | **85.61** | **9.86** | **174.93** |
| LIBERO | 2-STAGE | LAM30K + WM52K | 23.13 | 86.94 | 10.20 | 167.06 |
| | JOINT | WARM8K + E2E52K | **23.33** | **87.21** | **9.89** | **158.36** |

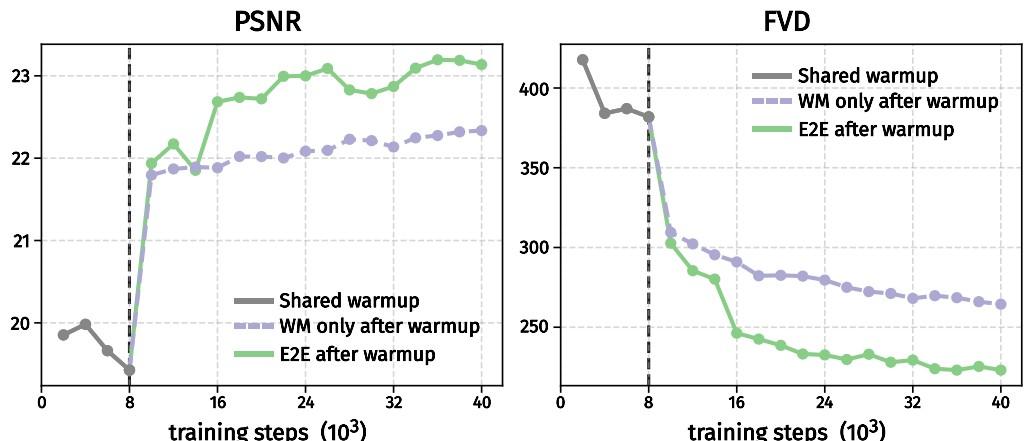

Figure 7: Complete WM video prediction performance curves during joint training, including both warm-up and end-to-end phases.

## D.2 ACTION TRANSFER RESULTS

Here we provide action transfer results in Figure 6, where our learned LAM in CoLA-World extracts the latent actions from the source video, and the world model generates the video from an initial image, taking these latent actions as conditions. For each video pair below, the top video is the source video, while the bottom one is the generated action-transfer video. We notice that the generated videos show a strong resemblance in semantic meaning to the source videos. To avoid too large PDF file, we provide additional qualitative results for action transfer videos in our anonymous repository.

Furthermore, we include comparative LAM transfer demos between our joint-trained CoLA-World and the two-stage baseline in our anonymous repository to show the benefits of joint training more clearly. Both models extract latent actions from the same source video and perform rollout imagination starting from the same initial target frame (taken from a dataset different from the source). The videos generated by the 2-stage baseline frequently fail to adhere to the semantic action of the source video, whereas CoLA-World faithfully preserves the intended behaviors. The comparison results highlight the superior semantic consistency and generation quality of our jointly trained model.

## D.3 PERFORMANCE OF LAM30K + WM52K

We conduct world model video prediction experiments for LAM30K+WM52K on datasets including OXE, EgoCentric, Agibot and Libero, same as Table 2. We relist the performance of Warm8K + E2E52K in Table 2 for easy comparison. The results are shown in Table 5. Warm8K + E2E52K

consistently matches or exceeds the performance of LAM30K + WM52K. This superiority again strongly validates the data efficiency of our joint training approach, which yields gains that cannot be achieved simply by increasing the budget of the two-stage pipeline.

### D.4 COMPLETE WM PREDICTION PERFORMANCE CURVES DURING JOINT TRAINING

We add new figures (Figure 7) illustrating the complete WM learning dynamics throughout warm-up and end-to-end training. Figure 7 extends the end-to-end WM performance curves in Figure 4(b) by including the preceding warm-up phase. During warm-up stage, the two performance curves overlap and remain low, reflecting that the World Model is frozen while the LAM aligns. This creates a stable foundation for the rapid improvements observed in the subsequent end-to-end phase.

### D.5 APPLICABILITY TO CONTINUOUS LATENT ACTIONS

Table 6: Video prediction performance of the learned world models on different datasets, where the used latent actions are continuous.

| DATASET | METHOD | | PSNR ↑ | SSIM ↑ | LPIPS ↓ | FVD ↓ |
|---|---|---|---|---|---|---|
| OXE | 2-STAGE | LAM30K + WM30K | 22.89 | 81.69 | 12.42 | 272.31 |
| | JOINT | WARM4K + E2E30K | 22.67 | 81.39 | 12.72 | 283.93 |
| EGOCENTRIC | 2-STAGE | LAM30K + WM30K | 24.67 | 84.75 | 11.70 | 238.46 |
| | JOINT | WARM4K + E2E30K | 24.50 | 84.56 | 11.93 | 241.56 |
| AGIBOT | 2-STAGE | LAM30K + WM30K | 24.41 | 86.18 | 9.62 | 169.45 |
| | JOINT | WARM4K + E2E30K | 24.23 | 86.00 | 9.78 | 172.51 |
| LIBERO | 2-STAGE | LAM30K + WM30K | 23.51 | 87.38 | 9.96 | 150.04 |
| | JOINT | WARM4K + E2E30K | 23.65 | 87.57 | 9.83 | 151.68 |

To demonstrate that our CoLA-World framework is agnostic to the latent format, we run additional experiments replacing the discrete VQ-quantizer with a continuous VAE bottleneck (similar to AdaWorld) while keeping the rest of the CoLA-World pipeline unchanged. We address the posterior collapse challenge often seen with strong generative backbones by enforcing a minimum KL threshold (kl_min=10). The weight of the KL loss in the VAE objective is 1e-3. We show WM video prediction performance of joint training and 2-stage paradigms in Table 6. The results demonstrate that our warm-up and joint training strategy successfully aligns the continuous LAM with the World Model, achieving a comparable performance with the 2-stage baseline with much higher data efficiency. This confirms that our co-evolutionary paradigm is robust and effective across both discrete and continuous latent actions.

### D.6 SCALABILITY

**Scalability with model size**: We hypothesize that since the World Model acts as a "tutor" providing gradients to the LAM, a larger, more capable WM should drive the LAM to learn superior representations. To validate this, we conducte a study that scales the WM (OpenSora) capacity by varying the number of used DiT blocks (6, 12, 20, and 28 blocks) during the E2E phase in CoLA-World training. We focus on results of the LAM quality (probing loss) since the WM video generation quality is expected to improve with larger models. As shown in Figure 8, we observe a distinct scaling trend: as the World Model size increases, the linear probing loss on LIBERO consistently decreases (improves). This confirms that CoLA-World effectively leverages the capacity of larger foundation models to learn a more expressive Latent Action Model.

**Scalability with training data size**: Following the protocol established by Genie (Bruce et al., 2024), we investigate the impact of training data scale by varying the batch size (64, 96, and 128). For each setting, we adjust the warm-up duration to ensure codebook utilization reaches a healthy, converged state (requiring 15K, 12K, and 8K steps, respectively), followed by end-to-end training for a fixed

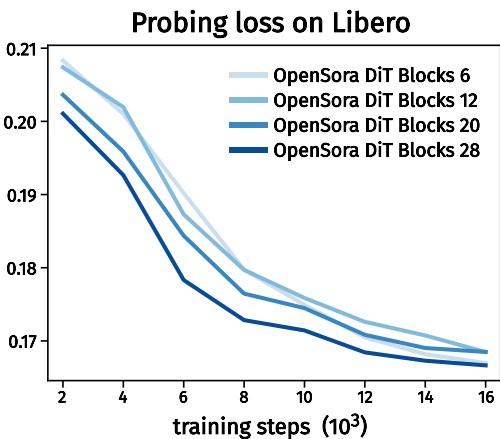

Figure 8: LAM probing curves on Libero during end-to-end training with different world model sizes (different numbers of DiT blocks used in the OpenSora model).

Table 7: LAM probing loss and the World Model's video prediction performance of CoLA-World training using different batch sizes.

| | WARMUP | E2E | PROBING LOSS ↓ | PSNR ↑ | SSIM ↑ | LPIPS ↓ | FVD ↓ |
|---|---|---|---|---|---|---|---|
| BATCH SIZE=64 | 15K | 20K | 0.1717 | 22.28 | 81.32 | 14.67 | 261.78 |
| BATCH SIZE=96 | 12K | 20K | 0.1687 | 22.44 | 82.04 | 13.46 | 249.32 |
| BATCH SIZE=128 | 8K | 20K | **0.1639** | **22.83** | **82.47** | **13.31** | **233.06** |

duration of 20K steps. We report the resulting LAM probing loss on LIBERO and the World Model's video prediction performance. As shown in Table 6 in our revised version, increasing the batch size—which corresponds to exposing the model to more data within the same training window—leads to consistent improvements in both LAM and WM performance.

