# OpenReview forum: "Co-Evolving Latent Action World Models"
_ICLR.cc/2026/Conference — Submitted to ICLR 2026_

### Official Review · Reviewer_nX3h · 2025-10-28

**Soundness:** 3
**Presentation:** 3
**Contribution:** 2
**Rating:** 6
**Confidence:** 4

**Summary:**

This paper proposes a joint training framework to learn a world model based on latent actions, including a warm-up phase where only the latent action model is optimized, and a joint-training phase. Experiments are conducted on several benchmarks to demonstrate the efficacy of the proposed framework.

**Strengths:**

- The paper is well-written. The proposed method is simple to follow, and I think it is easy to reproduce.
- The proposed method is sound. Given a well-pretrained world model, it is promising to learn both the latent action model and the world model jointly.
- The idea of adapting pre-trained video generative models to world models is promising and is an interesting topic for the community.
- Experiments include both seen and unseen datasets, which demonstrate the efficacy of the proposed method.

**Weaknesses:**

- The advantages of the proposed method are not obvious: **1) efficiency**: Can the proposed joint-training method save computation or accelerate training speed compared with the baseline? As the paper finetunes a large-scale pre-trained world model, the comparison of the required computation budget should be demonstrated clearly. **2) Applicability**: The proposed method requires a well-initialized, pre-trained world model for stable training. While the paper only studies Open-Sora, it is natural to ask, can this method work well for other models? If the pre-trained model is specifically worse than Open-Sora, it is challenging for the proposed method to perform well. **3) Scalability**: The baseline is scalable, which has been proven by Genie. Is the proposed method scalable with increased dataset size or model parameters?
- It is better to include qualitative results like video demos to show the performance improvements of the method. Moreover, I suggest including the video prediction performance at the warm-up stage and later joint-training stage for presenting the learning dynamics.
- Do the authors have any ideas for the data size used in the warm-up phase? Are there any monitoring metrics for better designing the warm-up phase to generalize to different training tasks and pre-trained models?

**Questions:**

Please see the weaknesses.

---

> ### Author Response · Authors · 2025-11-26
> **Response to Reviewer nX3h (Part 1)**
>
> We sincerely thank the reviewer for the positive assessment and constructive comments.
> We are pleased that you found the proposed CoLA-World framework "simple to follow" and "sound."
> We also appreciate your recognition that our experiments on both seen and unseen datasets effectively "demonstrate the efficacy" of the method.
> Please find below our detailed responses to the your concerns.
>
> > `W1.1:` efficiency: Can the proposed joint-training method save computation or accelerate training speed compared with the baseline? As the paper finetunes a large-scale pre-trained world model, the comparison of the required computation budget should be demonstrated clearly.
>
> `R1.1:` Yes, the proposed method can save wall-clock computation time.
> While the per-step cost is higher, the co-evolutionary synergy significantly accelerates convergence.
> As you suggest, we have provided a clear comparison of computation budgets in the general response, which demonstrates that our paradigm is efficient while unlocking superior downstream performance and robustness.
> Our method attains performance on par with the 2-stage baseline but requires fewer GPU-hours (~63h vs ~75h, ~100h vs ~110h).

---

> ### Author Response · Authors · 2025-11-26
> **Response to Reviewer nX3h (Part 2)**
>
> > `W1.3:` Scalability: The baseline is scalable, which has been proven by Genie. Is the proposed method scalable with increased dataset size or model parameters?
>
> `R1.3:` We appreciate you raising this insightful point regarding the scalability of our framework.
>
> * Scalability with Model Parameters: Yes, our method is scalable with increased model parameters. We hypothesized that since the world model acts as a "tutor" providing gradients to the LAM, a larger, more capable WM should drive the LAM to learn superior representations. To validate this, we conducted a study that scales the WM (OpenSora) capacity by varying the number of used DiT blocks (6, 12, 20, and 28 blocks) during the E2E phase in CoLA-World training. We focus on results of the LAM quality (probing loss) since the WM video generation quality is expected to improve with larger models. As shown in Figure 8 in our revised version, we observe a clear scaling trend: as the world model size increases, the linear probing loss on LIBERO consistently decreases (improves). This confirms that CoLA-World effectively leverages the capacity of larger foundation models to learn a more expressive Latent Action Model.
> * Scalability with Training Data Size: Following the protocol established by Genie, we investigate the impact of training data scale by varying the batch size (64, 96, and 128). For each setting, we adjust the warm-up duration to ensure codebook utilization reaches a healthy, converged state (requiring 15K, 12K, and 8K steps, respectively), followed by end-to-end training for a fixed duration of 20K steps. We report the resulting LAM probing loss on LIBERO and the world model's video prediction performance. As shown in Table 7 in our revised paper (or the same table below), increasing the batch size—which corresponds to exposing the model to more data within the same training window—leads to consistent improvements in both LAM and WM performance.
>
> | | WARMUP | E2E | Probing Loss $\downarrow$ | PSNR $\uparrow$ | SSIM $\uparrow$ | LPIPS $\downarrow$ | FVD $\downarrow$ |
> | :--- | :---: | :---: | :---: | :---: | :---: | :---: | :---: |
> | Batch Size=64 | 15K | 20K | 0.1717 | 22.28 | 81.32 | 14.67 | 261.78 |
> | Batch Size=96 | 12K | 20K | 0.1687 | 22.44 | 82.04 | 13.46 | 249.32 |
> | Batch Size=128 | 8K | 20K | **0.1639** | **22.83** | **82.47** | **13.31** | **233.06** |
>
> > `W1.2:` Applicability: The proposed method requires a well-initialized, pre-trained world model for stable training. While the paper only studies Open-Sora, it is natural to ask, can this method work well for other models? If the pre-trained model is specifically worse than Open-Sora, it is challenging for the proposed method to perform well.
>
> `R1.2:` We appreciate you raising this insightful point regarding the general applicability of our framework.
>
> We believe that the core principle of CoLA-World, namely addressing the "representation alignment gap" in joint LAM-WM training via a warm-up phase, is agnostic to the specific video generation architecture (e.g., OpenSora, Stable Video Diffusion[1], Wan[2]). The challenge of aligning a scratch LAM with a pre-trained backbone is universal, and our solution should transfer effectively. We will include the detailed results with alternative backbones in the final version of the paper. Regarding the concern about utilizing a less capable pre-trained model, our ablation study on model scaling (varying numbers of DiT blocks in OpenSora) serves as a direct empirical validation. By reducing the OpenSora model to use only 6 or 12 blocks, we effectively simulated "weaker" pre-trained backbones compared to the full 28-block model. As shown in Figure 8, the LAM probing loss for weaker models also consistently decrease throughout training, and the performance gap with the full model is relatively narrow by the late stages of E2E training. This demonstrates that a weaker pre-trained world model also successfully functions as a "tutor" to guide latent action learning, and the joint training mechanism remains effective across varying levels of model capability.

---

> ### Author Response · Authors · 2025-11-26
> **Response to Reviewer nX3h (Part 3)**
>
> > `W2:` It is better to include qualitative results like video demos to show the performance improvements of the method. Moreover, I suggest including the video prediction performance at the warm-up stage and later joint-training stage for presenting the learning dynamics.
>
> `R2:` Thank you for your suggestions. We have updated the anonymous repository (https://anonymous.4open.science/r/CoLA-World) to include comparative LAM transfer demos between our joint-trained CoLA-World and the two-stage baseline to show the benefits of joint training more clearly.
> Both models extract latent actions from the same source video and perform rollout imagination starting from the same initial target frame (taken from a dataset different from the source).
> Notably, the videos generated by the 2-stage baseline frequently fail to adhere to the semantic action of the source video, whereas CoLA-World faithfully preserves the intended behaviors.
> The comparison results highlight the superior semantic consistency and generation quality of our jointly trained model.
>
> As you suggest, we also add a new Figure 7 in the revised version illustrating the complete WM learning dynamics throughout warm-up and end-to-end training.
> Figure 7 extends the end-to-end WM performance curves in Figure 4(b) by including the preceding warm-up phase.
> During warm-up, the performance curve remains low, reflecting that the world model is frozen while the LAM aligns.
> This creates a stable foundation for the rapid improvements observed in the subsequent end-to-end phase.
>
> > `W3:` Do the authors have any ideas for the data size used in the warm-up phase? Are there any monitoring metrics for better designing the warm-up phase to generalize to different training tasks and pre-trained models?
>
> `R3:` We totally agree that determining the appropriate "data size" (or number of steps) for the warm-up phase is critical.
> As demonstrated in Figure 3, we rely on VQ Codebook metrics as the primary indicators to monitor the Warm-up process.
> Specifically, we track codebook utilization and entropy, where we look for the point where these metrics stop increasing and reach a plateau.
> We also track max-usage where we ensure this metric decreases and stabilizes, indicating that the model is avoiding "index collapse" (a few codes dominate).
> The warm-up phase is considered complete when these metrics converge/stabilize.
> In our experiments, this occurred around 8k-12k steps.
> We believe these representational stability metrics are applicable to different training tasks and pre-trained models.
> If the reviewer has specific alternative metrics in mind for other contexts, we would be very grateful for your suggestions to further enrich our discussion.
>
>
> [1] Blattmann, A., Dockhorn, T., Kulal, S., Mendelevitch, D., Kilian, M., Lorenz, D., Levi, Y., English, Z., Voleti, V., Letts, A. and Jampani, V., 2023. Stable video diffusion: Scaling latent video diffusion models to large datasets. arXiv preprint arXiv:2311.15127. \
> [2] Wan, T., Wang, A., Ai, B., Wen, B., Mao, C., Xie, C.W., Chen, D., Yu, F., Zhao, H., Yang, J. and Zeng, J., 2025. Wan: Open and advanced large-scale video generative models. arXiv preprint arXiv:2503.20314.

---

### Official Review · Reviewer_puks · 2025-10-29

**Soundness:** 1
**Presentation:** 3
**Contribution:** 2
**Rating:** 4
**Confidence:** 4

**Summary:**

This paper proposes a joint learning approach for the latent action model and latent action world model. It leverages a critical warm-up phase to avoid collapse. Such joint learning makes the world model stronger controllable and sample efficient.

**Strengths:**

* The problem that this paper would like to address is clear. Separately learning the latent action model and world model will hinder their in-depth mutual improvement.
* This paper conducts in-depth analysis of why direct joint learning is relatively difficult.
* The presentation is good for easy understanding.

**Weaknesses:**

* Lack of visualization comparison between the 2-stage and joint. The visualization results can better help to understand the benefits of joint learning.
* There is no significant improvement compared to the 2-stage for the performance of latent action or world models. It's very hard to see the benefit of joint learning
* Training budget is not the perfect metric to evaluate efficiency. The FDM model is much smaller than the world models.

**Questions:**

* What about the performance of LAM30K+WM52K?

---

> ### Author Response · Authors · 2025-11-26
> **Response to Reviewer puks**
>
> We sincerely thank the reviewer for the valuable feedback.
> We are glad that you found the problem of decoupled learning hindering mutual improvement to be "clear" and appreciated our "in-depth analysis" regarding the challenges of direct joint learning.
> Please find below our detailed responses to the your concerns.
>
> > `W1:` Lack of visualization comparison between the 2-stage and joint. The visualization results can better help to understand the benefits of joint learning.
>
> `R1:` Thank you for raising this point.
> We have updated the anonymous repository (https://anonymous.4open.science/r/CoLA-World) to include comparative LAM transfer demos between our joint-trained CoLA-World and the two-stage baseline to show the benefits of joint training more clearly.
>
> For each demo, we include three files: the source video, used to extract latent actions; CoLA-World generated video: video generated by the joint-trained CoLA-World's WM, conditioned on latent actions extracted by its corresponding jointly trained LAM's IDM; 2-Stage generated video: video generated by the two-stage trained WM, conditioned on latent actions extracted by its corresponding LAM's IDM trained in the separated first stage.
> Both models extract latent actions from the same source video and perform rollout imagination starting from the same initial target frame (taken from a dataset different from the source).
>
> Notably, the videos generated by the 2-stage baseline frequently fail to adhere to the semantic action of the source video, whereas CoLA-World faithfully preserves the intended behaviors. The comparison results highlight the superior semantic consistency and generation quality of our jointly trained model.
>
> > `W2:` There is no significant improvement compared to the 2-stage for the performance of latent action or world models. It's very hard to see the benefit of joint learning
>
> `R2:` We would like to clarify that the improvement of our method is systematic and substantial when evaluated holistically.
> Specifically, our CoLA-World paradigm demonstrates superior data efficiency (matching fully-trained 2-stage baselines with significantly less data), much higher downstream WM performance (achieving a clear performance gain on downstream real-action-based video prediction), enhanced LAM robustness (avoiding the representational collapse seen in 2-stage methods during adaptation), and higher utility (achieving a ~90% relative improvement in downstream planning success).
> Please refer to our general response for the detailed breakdown of these advantages.
>
> > `W3:` Training budget is not the perfect metric to evaluate efficiency. The FDM model is much smaller than the world models.
>
> `R3:` We agree that "training budget" (data steps) is not the only metric for efficiency given the model size differences.
> We have now conducted a rigorous wall-clock training time analysis to provide a complete picture.
> The results show that our method is in fact efficient in terms of both data and time, requiring fewer GPU-hours to match the fully-trained baseline (~63h vs ~75h, ~100h vs ~110h). Please refer to our general response for detailed runtime results and analysis.
>
> > `Q1:` What about the performance of LAM30K+WM52K?
>
> `R4:` Thank you for the question. During the rebuttal period, we conduct world model video prediction experiments for LAM30K+WM52K on datasets including OXE, EgoCentric, Agibot and Libero, same as Table 2. The results are shown in Table 5 in the revised paper or in the same table below. We relist the performance of  Warm8K + E2E52K in Table 2 for easy comparison. Warm8K + E2E52K consistently matches or exceeds the performance of LAM30K + WM52K. This superiority again validates the data efficiency of our joint training approach, which yields gains that cannot be achieved simply by increasing the budget of the two-stage pipeline.
> | Dataset | Paradigm | Method | PSNR $\uparrow$ | SSIM $\uparrow$ | LPIPS $\downarrow$ | FVD $\downarrow$ |
> | :--- | :--- | :--- | :---: | :---: | :---: | :---: |
> | **OXE** | 2-stage | LAM30K + WM52K | 22.54 | **81.45** | 12.87 | 281.05 |
> | | Joint | Warm8K + E2E52K | **22.57** | 81.40 | **12.79** | **278.90** |
> | **EgoCentric** | 2-stage | LAM30K + WM52K | **23.72** | **83.56** | **12.96** | 259.33 |
> | | Joint | Warm8K + E2E52K | 23.69 | 83.52 | 13.08 | **252.45** |
> | **AgiBot** | 2-stage | LAM30K + WM52K | 23.79 | 85.57 | 9.91 | 180.45 |
> | | Joint | Warm8K + E2E52K | **23.93** | **85.61** | **9.86** | **174.93** |
> | **LIBERO** | 2-stage | LAM30K + WM52K | 23.13 | 86.94 | 10.20 | 167.06 |
> | | Joint | Warm8K + E2E52K | **23.33** | **87.21** | **9.89** | **158.36** |

---

### Official Review · Reviewer_wCme · 2025-10-30

**Soundness:** 2
**Presentation:** 3
**Contribution:** 2
**Rating:** 4
**Confidence:** 4

**Summary:**

This paper proposes a latent action world model learning approach that jointly updates the latent action model (LAM) and the world model. The approach includes a warm-up phase designed to align the representations of the LAM and a pre-trained world model, after which both models are updated synergistically. This framework mitigates the representational collapse problem during joint learning of the LAM and world model, making it effective. Experimental results demonstrate that this joint learning approach produces a better latent action space, superior video simulation quality, and improved visual planning performance compared to a two-stage approach.

**Strengths:**

1.	The representational collapse problem addressed in this paper is well-motivated and thoroughly analyzed. The use of metrics such as utilization, maximum usage, and entropy to assess the quality of the latent space is intuitive and effective.
2.	The method proposed to address the representational collapse problem is simple, intuitive, and effective.

**Weaknesses:**

1.	The improvement in metrics such as linear probing loss and video prediction performance is marginal. It is difficult to accept that joint learning is definitively better.
2.	As shown in Fig. 2, the joint learning of a randomly initialized inverse dynamics model (IDM) and world model (WM) ultimately approaches the performance of a pre-trained IDM. This suggests that joint training from scratch may be a viable approach if a pre-trained video prediction model is unavailable, which contradicts the authors' claim of being the first successful joint training framework.
3.	More comparisons of video simulation quality and visual planning performance with other paradigms, such as joint training from scratch (PreLAR) or iterative learning (AD3), should be included and discussed.

**Questions:**

1.	In Fig. 2 and Fig. 3, the best metrics (such as utilization, maximum usage, and entropy) of the latent action space may be influenced by the distribution of actions in the training data. It would be beneficial to use normalized metrics or provide statistical results to demonstrate that the data is balanced.
2.	The authors use OpenSora as a pre-trained world model. Would using other types or architectures of pre-trained world models yield different results?

---

> ### Author Response · Authors · 2025-11-26
> **Response to Reviewer wCme (Part 1)**
>
> We sincerely thank the reviewer for the constructive feedback and encouraging assessment.
> We are gratified that you find the representational collapse problem "well-motivated and thoroughly analyzed," and our proposed solution "simple, intuitive, and effective".
> Please find below our detailed responses to your concerns.
>
> > `W1:` The improvement in metrics such as linear probing loss and video prediction performance is marginal. It is difficult to accept that joint learning is definitively better.
>
> `R1:` We would like to clarify that the improvement of our method is systematic and substantial when evaluated holistically.
> Specifically, our CoLA-World paradigm demonstrates superior data efficiency (matching fully-trained 2-stage baselines with significantly less data), much higher downstream WM performance (achieving a clear performance gain on downstream real-action-based video prediction), enhanced LAM robustness (avoiding the representational collapse seen in 2-stage methods during adaptation), and higher utility (achieving a ~90% relative improvement in downstream planning success).
> Please refer to our general response for the detailed breakdown of these advantages.
>
> > `W2:` As shown in Fig. 2, the joint learning of a randomly initialized inverse dynamics model (IDM) and world model (WM) ultimately approaches the performance of a pre-trained IDM. This suggests that joint training from scratch may be a viable approach if a pre-trained video prediction model is unavailable, which contradicts the authors' claim of being the first successful joint training framework.
>
>
> `R2:` We appreciate the reviewer's detailed examination of Figure 2.
> However, we would like to clarify that the results of IDM(rand) + WM(rand) (training from scratch) does not contradict our claim, nor does it represent a superior solution, for the following reasons:
>
> * The primary motivation is to leverage the vast knowledge embedded in video foundation models (like OpenSora), rather than discarding it. Training such a massive model (1.2B parameters) entirely from scratch as a mere FDM is computationally inefficient, and is a degenerate solution that yields inferior codebook quality compared to our method (Figure 3). The IDM(rand) + WM(rand) variant in Figure 2 is only intended to motivate our warm-up + end2end training strategy.
> * Our claim is specific to the context of "adapting pre-trained video generation models". The core challenge we address is the representation gap that causes collapse when combining a from-scratch LAM with a fixed pre-trained video generation model. IDM(rand) + WM(rand) reduces the setup to a standard LAM training pipeline (with an enormously larger FDM). It may or may not eventually approach the 2-stage baseline (the dashed line), but this is tangential to our primary research objective. In contrast, our proposed method is the first to successfully enable joint training while retaining and leveraging the pre-trained backbone.

---

> ### Author Response · Authors · 2025-11-26
> **Response to Reviewer wCme (Part 2)**
>
> > `W3:` More comparisons of video simulation quality and visual planning performance with other paradigms, such as joint training from scratch (PreLAR) or iterative learning (AD3), should be included and discussed.
>
> `R3:` We thank the reviewer for highlighting PreLAR and AD3.
> We will expand our related work section to discuss these methods in detail, but we would like to clarify that a quantitative performance comparison is not applicable due to fundamental differences in problem settings and research goals:
>
> AD3 (Iterative Learning for Online Visual RL): AD3 focuses on an online RL setting to distinguish "implicit distractor actions" from task-relevant ones using a latent action model. It iteratively trains the latent action model and uses the newly updated distractor action to train the separate world models, using the online collected data. Crucially, its Forward Dynamics Model is typically a lightweight model trained from scratch, rather than a generative foundation model. Its world model is also a lightweight Dreamer-based one. Its primary setting (handling distractors in online RL) differs from ours (leveraging pre-trained video priors to train a latent-action-based WM).
>
>
> PreLAR (Joint Training from Scratch): PreLAR also aims to train a world model using in-the-wild video data and also involves joint training of a latent action model with the world model.
> However, it trains both the LAM and the world model (a lightweight Dreamer-based one) entirely from scratch.
> It does not face the specific "representation alignment" challenge that arises when integrating a randomly initialized LAM with a highly structured, pre-trained video generative model.
>
> In contrast, our work is specifically motivated by the goal of jointly training a latent-action-based world model leveraging powerful pre-trained video foundation models.
> Our core contribution is solving the "alignment gap" and "representation collapse" that occur when a from-scratch LAM meets a pre-trained backbone.
> Consequently, a direct performance comparison between our paradigm and from-scratch or online-iterative methods is less meaningful.
> We will expand our related work section to deeply discuss these methodological differences to better position our contribution.
>
> > `Q1:` In Fig. 2 and Fig. 3, the best metrics (such as utilization, maximum usage, and entropy) of the latent action space may be influenced by the distribution of actions in the training data. It would be beneficial to use normalized metrics or provide statistical results to demonstrate that the data is balanced.
>
> `R4:` We appreciate the reviewer's rigorous thinking regarding the influence of data distribution on codebook metrics.
> We agree that the values of utilization and entropy can be influenced by the underlying distribution of actions in the dataset.
> However, we would like to clarify that data balance is not the cause of the performance gaps observed in Figures 2 and 3, for the following reasons:
>
> * All experiments in Figures 2 and 3 were conducted using the exact same training dataset mixture, and the data distribution is a constant factor across all comparisons. The drastic divergence in codebook metrics, where the baseline collapses to near-zero utilization (gray line in Figure 2) while our method maintains high entropy and utilization ("8k warm-up steps" in Figure 3), is a result of the training dynamics (e.g., representation alignment via warm-up) rather than the data distribution itself.
> * While natural data imbalance might lead to non-uniform usage, it differs from the representational collapse observed in the baselines, where the model degenerates to using only a handful of codes for all inputs. CoLA-World effectively prevents this failure mode, enabling the model to utilize the codebook capacity effectively despite the potential imbalance of the data distribution.
>
> > `Q2:` The authors use OpenSora as a pre-trained world model. Would using other types or architectures of pre-trained world models yield different results?
>
> `R5:` Thank you for raising this point.
> We believe that the core principle of CoLA-World, namely addressing the "representation alignment gap" in joint LAM-WM training via a warm-up phase, is agnostic to the specific video generation architecture (e.g., OpenSora Stable Video Diffusion[1], Wan[2]). We will include the detailed results with alternative backbones in the final version of the paper.
>
>
> [1] Blattmann, A., Dockhorn, T., Kulal, S., Mendelevitch, D., Kilian, M., Lorenz, D., Levi, Y., English, Z., Voleti, V., Letts, A. and Jampani, V., 2023. Stable video diffusion: Scaling latent video diffusion models to large datasets. arXiv preprint arXiv:2311.15127. \
> [2] Wan, T., Wang, A., Ai, B., Wen, B., Mao, C., Xie, C.W., Chen, D., Yu, F., Zhao, H., Yang, J. and Zeng, J., 2025. Wan: Open and advanced large-scale video generative models. arXiv preprint arXiv:2503.20314.

---

### Official Review · Reviewer_dyxF · 2025-11-01

**Soundness:** 3
**Presentation:** 4
**Contribution:** 3
**Rating:** 6
**Confidence:** 5

**Summary:**

The paper aims to improve the pretraining of the generalist action-conditioned world models. To achieve that, it proposes to replace the forward dynamics model in previous literature with the world model, and jointly pretrain the latent action model and the world model. To avoid representation collapse, the paper further designs a warm-up stage. The paper conducts downstream applications in multiple decision making scenarios in terms of video prediction quality and visual planning performance.

**Strengths:**

S1) The paper presents a very detailed and inspiring analysis of different integration methods of latent action model, which I believe will be highly instructive to the readers.

S2) The proposed method is well-motivated, simple, and effective.

S3) The proposed method is validated on multiple important settings, demonstrating its superiority.

**Weaknesses:**

W1) The paper is not the first practice to replace FDM in the latent action model with a generative world model. UniSkill [1] has already explored this setting, which is missing in the discussion of the paper.

W2) My major concern is the training efficiency of the proposed method. Previous two-stage methods can enjoy a high training throughput by using a light-weighted FDM. However, this paper uses OpenSora, which is more computational heavy. It is important to consider the training efficiency in comparison.

W3) Another concern is about the expressiveness of the jointly learned latent actions. As shown in Figure 2, "IDM (rand) + WM (rand)" has better codebook usage compared with "IDM (pre) + WM (pre)". In my opinion, "IDM (rand) + WM (rand)" is similar with the case of the two-stage method, which trains IDM and FDM from scratch, while "IDM (pre) + WM (pre)" is the proposed method. Hence, does that indicate that prior method learns a latent action codebook with larger capcacity? As the pretrained WM exists some prior knowledge, it may undermine the learning of an expressive codebook. On the contrary, the two-stage method could motivate the learning of codebook to represent more action information since the FDM has limited prediction capability.

---

[1] UniSkill: Imitating Human Videos via Cross-Embodiment Skill Representations

**Questions:**

Q1) In AdaWorld, it is demonstrated that a continuous latent action space can perform better than discrete latent actions in application. Why this paper doesn't consider to use continuous latent actions?

---

> ### Author Response · Authors · 2025-11-26
> **Response to Reviewer dyxF (Part 1)**
>
> We sincerely thank the reviewer for the thoughtful feedback and positive assessment.
> We are particularly encouraged by your recognition of our "detailed and inspiring analysis" regarding different integration methods, as well as your assessment that our proposed CoLA-World is "well-motivated, simple, and effective".
> Please find below our detailed responses to the your concerns.
>
>
> > `W1:` The paper is not the first practice to replace FDM in the latent action model with a generative world model. UniSkill [1] has already explored this setting, which is missing in the discussion of the paper.
>
> `R1:` We thank the reviewer for pointing out this related work. While UniSkill shares the similar idea of using a generative model as the FDM in LAM learning, CoLA-World differs significantly in objective and mechanism.
>
> * The generative FDM (InstructPix2Pix) used in UniSkill remains an auxiliary module in latent action (skill) learning and is discarded after training. It is not used for downstream simulation or planning. In contrast, our CoLA-World focuses on building a latent-action-based world model using a pretrained generative video foundation model (e.g., OpenSora). Here the world model (FDM) itself is one of the primary products, intended for video simulation (Table 2 & 3) and visual planning (Table 4).
> * UniSkill formulates the learning of FDM as an image editing task using InstructPix2Pix, rather than a world model or a video generation model. UniSkill predicts the next frame by modifying the current frame based on a latent action (skill). In CoLA-World, we directly use a video generation model (OpenSora) as the FDM, so that we can finetune the video foundation model into a controllable world model with the jointly trained LAM.
>
> We will cite this paper and include detailed discussion on how our work relates to and differs from theirs in the final version.
>
>
>
> > `W2:` My major concern is the training efficiency of the proposed method. Previous two-stage methods can enjoy a high training throughput by using a light-weighted FDM. However, this paper uses OpenSora, which is more computational heavy. It is important to consider the training efficiency in comparison.
>
> `R2:` Thank you for raising this important point.
> You are right that the 2-stage method has higher per-step throughput due to the lightweight FDM.
> However, as detailed in our general response, the training wall-clock results show that our synergistic evolution accelerates training convergence, and outweighs the per-step cost.
> Our method is actually more time-efficient than the 2-stage baseline (~63h vs ~75h, ~100h vs ~110h).
> Please refer to the general response for the detailed GPU-hour comparison.

---

> ### Author Response · Authors · 2025-11-26
> **Response to Reviewer dyxF (Part 2)**
>
> > `W3:` Another concern is about the expressiveness of the jointly learned latent actions. As shown in Figure 2, "IDM (rand) + WM (rand)" has better codebook usage compared with "IDM (pre) + WM (pre)". In my opinion, "IDM (rand) + WM (rand)" is similar with the case of the two-stage method, which trains IDM and FDM from scratch, while "IDM (pre) + WM (pre)" is the proposed method. Hence, does that indicate that prior method learns a latent action codebook with larger capcacity? As the pretrained WM exists some prior knowledge, it may undermine the learning of an expressive codebook. On the contrary, the two-stage method could motivate the learning of codebook to represent more action information since the FDM has limited prediction capability.
>
> `R3:` We would like to clarify that the variants shown in Figure 2 are primarily intended to motivate our method.
> Although IDM (rand) + WM (rand) shares some similarity with the two-stage approach in that both training the IDM and FDM from scratch, IDM (pre) + WM (pre) is **not** our proposed joint learning method.
> Our CoLA-World employs end-to-end training following a warm-up phase, i.e. IDM(warmup) + WM(pre) (as shown in Figure 3).
> In contrast, IDM(pre) + WM(pre) initializes the IDM in joint training using weights trained with a lightweight FDM (same as in the two-stage approach).
> Its poor codebook metrics in Figure 2 proves that the IDM learned with an external FDM is misaligned with the powerful representations of the pre-trained world model.
>
> We agree with your insight that a pre-trained WM may undermine the learning of a scratch codebook—this is precisely the collapse shown in the "IDM (rand) + WM (pre)" curve in Figure 2, which motivated our warm-up strategy to align representations so the WM can guide rather than undermine the latent action learning.
>
> Furthermore, while IDM(rand) + WM(rand) avoids total collapse compared to other misaligned variants in Figure 2, its codebook quality still significantly lags behind our CoLA-World.
> Comparing Figure 2 with Figure 3 confirms that our CoLA-World method achieves much higher utilization and entropy.
> This confirms that leveraging the rich prior knowledge of a pre-trained video generative model (as a powerful FDM) yields a far more expressive latent space than training everything from scratch.
>
> > `Q1:` In AdaWorld, it is demonstrated that a continuous latent action space can perform better than discrete latent actions in application. Why this paper doesn't consider to use continuous latent actions?
>
> `R4:` Thank you for pointing this out.
> We chose discrete latent actions based on VQ primarily to align with standard baselines in the field (e.g., Genie).
> However, we emphasize that the CoLA-World framework (Joint Training with Warm-up) is agnostic to the specific format of the latent bottleneck and is applicable to continuous latent action spaces (e.g., VAE-based).
>
> To demonstrate this generality, we run additional experiments replacing the discrete VQ-quantizer with a continuous VAE bottleneck (similar to AdaWorld) while keeping the rest of the CoLA-World pipeline unchanged. We address the posterior collapse challenge often seen with strong generative backbones by enforcing a minimum KL threshold (kl\_min=10). The weight of the KL loss in the VAE objective is 1e-3. We show WM video prediction performance of joint training and 2-stage paradigms in Table 6 in the revised paper. The results demonstrate that our warm-up and joint training strategy successfully aligns the continuous LAM with the World Model, achieving a comparable performance with the 2-stage baseline with much higher data efficiency. This confirms that our co-evolutionary paradigm is robust and effective across both discrete and continuous latent actions.

---

### Author Response · Authors · 2025-11-26

We sincerely thank all reviewers for their insightful feedback and encouraging assessment.
We are excited to see that the reviewers recognized our work as "well-motivated" (R1, R2, R3, R4) and found the proposed CoLA-World framework to be "simple, intuitive, and effective" (R1, R2, R4).
Reviewers particularly appreciated our "detailed and inspiring analysis" of the representational collapse problem (R1, R2, R3).

We have identified two major recurring concerns raised by multiple reviewers: (1) **computational efficiency** and (2) **the magnitude of performance improvements**.
To address these comprehensively, we provide a general response here, followed by detailed individual point-by-point replies.

---

> ### Author Response · Authors · 2025-11-26
>
> (1) **General Response to Computational Efficiency**:
>
> We conduct all experiments using 8 NVIDIA H200 GPUs.
> For the 2-stage training paradigm, the 30K-step LAM training takes about 30 hours (about 3.6s / step), while training the world model on the fixed LAM for 30K steps takes about 45 hours (about 5.5s / step).
> For our joint training paradigm, the whole process (Warm8K + E2e52K) takes about 100 hours (about 6s / step).
> The total training times for key runs are listed below:
> |  Setting | Method | Total Data Steps | Total Training Time|
> | :--- | :--- | :---: | :---: |
> | 2-Stage | LAM30K + WM30K | 60K | ~75h |
> | 2-Stage (Long) | LAM30K + WM52K | 82K | ~110h |
> | Joint | WARM8K + WM30K| 38K | ~63h |
> | Joint (Complete) | WARM8K + WM52K| 60K | ~100h |
>
> We acknowledge that the per-step computation cost of the joint paradigm is slightly higher than the 2-stage approach due to the overhead of backpropagating through the large world model (OpenSora) compared to training a lightweight FDM.
>
> However, as shown in Table 2 of the paper, our WARM8K+E2E30K model already rivals the fully trained 2-stage baseline (LAM30K+WM30K). Crucially, this level of performance is achieved in \~63 hours, compared to \~75 hours for the 2-stage baseline. When training is extended to 60K steps (WARM8K+E2E52K), we achieve performance (see Table 5) that slightly outperforms 2-stage baseline (WARM30K+E2E52K), even when the baseline is trained for longer (\~100h v.s \~110h). These strong evidence clearly demonstrates that the co-evolutionary synergy accelerates convergence, allowing us to efficiently reach target performance **using both less data and fewer GPU-hours**.
>
> Moreover, our joint training method also brings the following advantages:
>
> * The speed of the LAM trained using a lightweight FDM comes at the cost of learning a "shortcut" representation, which proves fragile during downstream adaptation. Figure 5 shows that the 2-stage method suffers from representational collapse when adapting to real actions. In contrast, CoLA-World maintain a relatively healthy latent space. This computation investment yields superior downstream real-action-based video prediction performance and visual planning success rate.
> * Our paradigm removes the redundancy of the 2-stage approach, which trains a FDM that is discarded after LAM training. CoLA-World ensures that every training FLOP contributes directly to the final deployable world model.
>
> Furthermore, our main contribution is establishing the paradigm for jointly training latent actions with a pre-trained video-generation-based world model. Our priority is to validate the feasibility and superior effectiveness of this co-evolutionary framework.
> With this paradigm now established, we believe that further computational optimizations (e.g., efficient model quantization) are complementary directions to our work, rather than a limitation of the current contribution.

---

> ### Author Response · Authors · 2025-11-26
>
> (2) **General Response to the Magnitude of Performance Improvements**:
>
> We argue that evaluating the improvement solely based on the absolute performance on probing or video prediction in Section 4.2 provides an incomplete picture.
> We believe our performance advantage is systematic and substantial when the LAM and world model are evaluated holistically across efficiency, adaptability, robustness and utility:
> * Data Efficiency: While the absolute performance gap in Table 2 might appear modest, the cost to achieve it is significantly lower. Our WARM8K + E2E30K model (using nearly half the training budget of the fully-trained 2-stage baseline LAM30K + WM30K) already matches or outperforms the 2-stage baseline across different datasets. This is also the case when comparing WARM8K+E2E52K with LAM30K+WM52K (Table 5). This proves that our joint training paradigm exploits data interactions much more effectively than the 2-stage approach.
> * WM Performance Improvement in Downstream Adaptation: The most significant gains appear when the model faces the real-world challenge of adapting to real action spaces (Section 4.4). In Table 3 (Real Action), CoLA-World achieves a clear improvement over the 2-stage baseline (e.g., FVD 93.68 vs. 115.45 on LIBERO, LPIPS 8.90 vs. 10.64 on RoboDesk). This is not marginal; it is a qualitative leap.
> * LAM Robustness in Downstream Adaptation: As analyzed in Figure 5, the real-action-based WM performance gain stems from the fact that our jointly trained LAM is intrinsically robust. The 2-stage LAM suffers from "adapter collapse" (utilizing only ~10% of the codebook) during adaptation, whereas CoLA-World maintains a relative healthy, diverse representation. Crucially, these results in Figure 5 offers a deeper perspective on the "marginal" linear probing differences observed in Table 1, suggesting that linear probing loss captures only one aspect of representational quality. The adaptation task effectively uncovers the latent fragility of the 2-stage baseline, and the superiority of our jointly learned LAM are more evidently revealed by its resilience in practical downstream adaptation.
> * Superior Utility in Planning: Ultimately, a world model is a tool for control. In the VP2 planning benchmark (Table 4), our method nearly doubles the relative success rate compared to the 2-stage baseline. This demonstrates that our improved simulation quality is not just a statistical artifact but translates into reliable, actionable predictions for downstream planners.

---

### Author Response · Authors · 2025-12-03
**Summary of Reviews and Rebuttal for the Area Chair**

Dear Area Chair,

Thank you for overseeing the review process. We would like to provide a brief factual summary of our contributions and the updates made during the rebuttal period.

**Summary of Contributions:**
We propose CoLA-World, the first framework that successfully enables joint training of a latent action model with a pre-trained video-generation-based world model. Compared to prior two-stage methods, CoLA-World’s joint latent action learning and world modeling yield a higher-quality latent action space and a world model with stronger controllability, improving both video simulation and downstream visual planning. We show that CoLA-World’s joint training exhibits synergistic co-evolution: the improving world model and LAM mutually reinforce each other, creating a tightly coupled system that drives superior adaptability.

**Overall Assessment from Reviewers**:
The reviewers collectively endorsed the significance and promise of our core research direction—jointly training latent action models with world models—recognizing the work as "well-motivated" (R1, R2, R3, R4) and "sound" (R4). They consistently found the proposed CoLA-World framework to be "simple, intuitive, and effective" (R1, R2, R4) as well as "easy to reproduce" (R4). Reviewers particularly appreciated our "detailed and inspiring", "thorough" and "in-depth" analysis of the difficulty of the representational collapse problem in joint learning, noting it would be "highly instructive" to readers (R1, R2, R3). Furthermore, the reviewers acknowledged the rigorous validation across multiple important settings, demonstrating the "efficacy", "soundness" and  "superiority" of our approach (R1, R4).

**Key Concerns & Our Rebuttal Actions**:

* Computational Efficiency: Since our method involves backpropagation through a large World Model (OpenSora), reviewers (R1, R3, R4) were concerned that it might be computationally heavier and less efficient than the 2-stage baseline (which trains a lightweight FDM).

    We conducted a rigorous evaluation on the computation cost and the results (provided in the General Response) show that our method is actually more time-efficient. Our WARM8K+E2E30K / WARM8K+E2E52K model achieves comparable or better performance than the 2-stage baseline (LAM30K+WM30K / LAM30K+WM52K). This proves that our synergistic acceleration of co-evolution outweighs the per-step computational cost, our method is both data- and computation-efficient.

* Magnitude of Performance Improvement: Some reviewers (R2, R3) felt our performance improvement over the two-stage baseline was "marginal".

    We clarified that the performance improvement is systematic and substantial when evaluated holistically. Our method demonstrates much higher data efficiency, matching baseline performance with significantly less data. Our joint-learned latent action model is much more robust than the 2-stage baseline during downstream adaptation, avoiding the potential representational collapse. The joint-learned world model also has significant performance gains in downstream real-action-based video prediction and visual planning.


* Scalability & General Applicability: Some reviewers have concerns on the proposed method's scalability (R4) and applicability to different latent settings (R1).

    Following the evaluating protocol in Genie, we provided new experiments showing that the performance improves consistently with larger world model sizes (varying DiT blocks) and larger training data size (different batch sizes), highlighting our method's scalability. We also validated that our CoLA-World paradigm is agnostic to the latent format, successfully demonstrating its effectiveness with continuous latent actions (VAE-based) in addition to the discrete (VQ) setting used in the main paper.


We have addressed reviewer concerns with new experiments, visualizations, and clear explanation of our method. We hope the revisions and responses make the significance and impact of our work clear.

Best regards,\
Authors

---

### Meta-Review · Area_Chair_m5bx · 2026-01-06

**Summary:**

This paper proposes to jointly learn a latent action encoder with a world model, allowing latent actions to coevolve with the training of the video generation model. The authors illustrate how directly training the latent action encoder leads to training collapse as the pretrained video model learns to ignore the latents generated by the encoder. The authors propose a new cotraining method, where the video model weights are initially frozen and the latent action trained before both are jointly finetuned.

The reviewers were split on the paper. Reviewers noted that the paper was well motivated and simple. However, there were concerns about the novelty of the paper, as prior work such as Uniskill have already explored joint training of the latent action model. In addition, the performance gains of the method are rather marginal, and the AC after inspecting the uploaded sample videos could not tell a significant difference between the jointly trained video model compared to the non jointly trained one. In addition, when the AC was reading the paper, some of the details of the approach are not adequetly discussed, for instance it is not clear how the latent action world model is adapted to the planning setting.

Based off the these remaining concerns, the AC recommends rejecting the paper.

**Reviewer Concerns:**

Reviewers had some concerns about the computational efficiency of the method that were well addressed. However, there were outstanding concerns about the novelty of the paper, as prior work such as Uniskill have already explored joint training of the latent action model. In addition, the outstanding concern about the performance gains of the method are rather marginal are maintained, and the visual comparisons between both joint or seperate models is very limited.

**Reviewer Scores:**

I believe the rebuttal did not sufficiently address each of the reviewers concerns. Reviewer dyxF and  nX3h are likely to maintain a boarderline acceptance score while Reviewer wCme and puks are likely to maintain a boarderline reject scores.

---

### Decision · Program_Chairs · 2026-01-26

Reject